# Surveillance of Chemical Foodborne Disease Outbreaks in Zhejiang Province, China, 2011–2023

**DOI:** 10.3390/foods14060936

**Published:** 2025-03-10

**Authors:** Lili Chen, Jiang Chen, Jikai Wang, Xiaojuan Qi, Hexiang Zhang, Yue He, Ronghua Zhang

**Affiliations:** Department of Nutrition and Food Safety, Zhejiang Provincial Center for Disease Control and Prevention, 3399 Binsheng Road, Binjiang District, Hangzhou 310051, China; llchen@cdc.zj.cn (L.C.); jchen@cdc.zj.cn (J.C.); jkwang@cdc.zj.cn (J.W.);

**Keywords:** foodborne disease outbreak, chemical, nitrite, lead

## Abstract

Foodborne diseases are a growing public health problem worldwide, and chemical foodborne disease outbreaks (FBDOs) often have serious consequences. This study aimed to explore the epidemiological characteristics of chemical FBDOs in Zhejiang Province, China, and propose targeted prevention and control measures. Descriptive statistical methods were used to analyze chemical FBDO data collected from the Foodborne Disease Outbreaks Surveillance System in Zhejiang Province from 2011 to 2023. From 2011 to 2023, 74 chemical FBDOs were reported in Zhejiang Province, resulting in 461 cases, 209 hospitalizations, and one death. In contrast to other types of FBDOs, the percentage of hospitalized cases in chemical FBDOs was the highest (45.34%) (chi-square = 1047.9, *p* < 0.001). Outbreaks caused by nitrite accounted for the largest percentage (56.76%), followed by lead (17.57%). Outbreaks caused by nitrite occurred mainly in households (27), followed by restaurants (6), street stalls (5), and work canteens (3). Among all nitrite-related outbreaks, 59.52% (25/42) were caused by cooking food where it was used as a common seasoning, 26.19% (11/42) by eating pickled vegetables, 7.14% (3/42) by eating cooked meat products, and 4.76% (2/42) by eating grain products. Outbreaks caused by the misuse of nitrite in cooking mainly occurred in households (68%, 17/25), street stalls (16%, 4/25), work canteens (8%, 2/25), and restaurants (8%, 2/25). Outbreaks caused by eating pickled vegetables occurred mainly in households (90.91%, 10/11), and one outbreak occurred in a work canteen. Outbreaks caused by lead (n = 13) occurred in households, and liquor was involved in 12 outbreaks where they were caused by residents consuming yellow rice wine stored in tin pots. In view of the frequent outbreaks of chemical foodborne diseases in our province from 2011 to 2023, a variety of prevention and control measures were proposed based on the research results of the temporal and regional distribution, food and food establishments involved, and the etiological agents of the chemical FBDOs. However, the effectiveness of these recommendations needs to be further verified and studied. In general, public health institutions should further strengthen the surveillance and health education of the population. Individuals should store toxic chemicals, such as nitrates, pesticides, and rodenticides correctly to avoid poisoning by ingestion. In view of the chemical FBDOs caused by food in the catering and distribution links, relevant departments should strengthen targeted supervision.

## 1. Introduction

Foodborne diseases are usually infectious or toxic in nature and caused by bacteria, viruses, parasites, or chemical substances entering the body through contaminated food [1]. Foodborne diseases are a growing public health concern worldwide. Every year, 600 million people worldwide fall ill from eating contaminated food, resulting in 420,000 deaths and 33 million healthy life years lost [2]. Children are particularly affected, with 125,000 children under five dying each year [2]. In low- and middle-income countries, the total productivity loss associated with foodborne diseases is estimated to be USD 95.2 billion per year, and the annual cost of treating foodborne diseases is estimated at USD 15 billion [3].

Chemical foodborne disease is an important type of foodborne disease, which refers to a group of diseases caused by toxic chemicals that enter the human body through the ingestion of food or drinks. Chemical contamination can lead to acute toxicity or long-term diseases such as cancer. Common causes of chemical foodborne diseases include eating food contaminated with toxic chemicals, mistakenly eating toxic chemicals as food or as food additives, and eating food with excessive amounts of food additives. The symptoms of foodborne diseases caused by chemical contaminants range from mild gastroenteritis to fatal hepatic, renal, and neurological syndromes [4]. In 2010, the World Health Organization estimated that the disease burden of just four chemicals was associated with 339,000 diseases, 20,000 deaths, and 1,012,000 disability-adjusted life years. These should be considered the “tip of the iceberg” in terms of foodborne chemicals and their impact on the global burden of disease [2]. Minamata disease is caused by methylmercury poisoning that first occurred, at a large scale, in Minamata and neighboring communities in Japan in the 1950s and the 1960s [5,6] from the consumption of fish and shellfish in river water contaminated with organic mercury. Patients presented with neurological symptoms including paresthesia, ataxia, constriction of the visual field, dysarthria, and hearing difficulties [7]. At that time, many children in the exposed area were born with conditions similar to cerebral palsy [8], later known as congenital Minamata disease, having been affected in utero by methylmercury during exposure. In Japan, methylmercury contamination has resulted in 2263 government-confirmed adult cases and 63 cases of congenital Minamata disease, and 1368 patients with acute Minamata disease and 13 with congenital Minamata disease have died [9]; the effects of this disaster are still felt today. In 1971–1972, 459 deaths occurred in Iraq after grain treated with a methyl mercury fungicide was inadvertently used to make bread instead of being planted [10]. In 1985, California and other western states in the US experienced the largest outbreak of foodborne pesticide illness ever recorded in North America, resulting in 1376 cases of watermelon contamination by farmers using aldicarb [11]. In 1995, a chemical FBDO in India resulted in 22 cases and 14 deaths. Epidemiological data show that the outbreak was caused by the accidental use of sodium nitrite and potassium arsine to replace salt in the preparation of Tamarindus soup [12]. In 2005, approximately 50 men in rural Sri Lanka consumed kasippu (an illicit alcoholic beverage) contaminated with paraquat while attending a funeral, and 5 developed persistent progressive dyspnea and died 9–30 days after exposure [13]. In 2013, an insecticide food poisoning outbreak in India killed 23 children, with more than 48 requiring treatment [14]. The poisoning was caused by the consumption of a free lunch consisting of rice, soybeans, and lentils cooked with oil contaminated with an organophosphorus pesticide. As international trade increases and food chains become longer and more complex, the risk of food contamination and the movement of infected food across national borders increases. One study showed that consumption of food imported from Oaxaca was a risk factor for an elevated blood lead level in people in Monterey County, California [15]. In addition to large outbreaks that can cause serious health damage or economic losses, small outbreaks of foodborne chemical diseases often occur within households. Two families reported children poisoned by lead-contaminated spices purchased abroad, brought to the United States, and used to prepare their food [16]. In severe cases, these outbreaks can destroy small families. For example, a family of five was poisoned by lead-contaminated flour, resulting in two deaths [17]. Therefore, it is necessary for countries or regions to conduct surveillance programs on chemical FBDOs and understand their epidemiological characteristics to lay the foundation for the early identification and formulation of effective response strategies.

To detect and warn of FBDOs early, and to avoid major outbreaks, the World Health Organization recommends that countries establish food safety management frameworks and prioritize foodborne disease surveillance. In 2011, China established a web-based foodborne disease surveillance platform that has incrementally played a role in the early warning of food safety, food safety emergencies, and foodborne disease burden research [18]. The Foodborne Disease Outbreak Surveillance System (FDOSS), one of the surveillance systems on the platform, collects FBDOs with two or more cases as well as FBDOs with one or more deaths. These outbreaks are reported to the FDOSS after epidemiological investigations by local Centers for Disease Control and Prevention (CDCs), and most of the pathogenic factors, food vehicles, settings, and other information can be determined. As the first province to participate in foodborne disease outbreak surveillance, Zhejiang Province has accumulated a large amount of outbreak surveillance data. After more than ten years of effort, the timeliness and reporting rates of FBDO responses have significantly improved. By analyzing outbreak surveillance data, the common pathogenic factors, high-risk settings, and high-risk food vehicles that cause FBDOs can be determined. This study aimed to analyze the data of chemical FBDOs reported to the FDOSS in Zhejiang Province from 2011 to 2023 and provide a scientific basis for government to formulate and modify prevention and control strategies.

## 2. Methods

### 2.1. Definitions

An FBDO occurs when two or more individuals develop similar illnesses after ingesting the same contaminated food or drink [19]. In some countries, only one case of a rare but severe foodborne disease, such as botulism or chemical intoxication, is considered an outbreak [20]. According to the National Foodborne Disease Surveillance Manual, the FDOSS in China collects FBDOs with two or more cases as well as FBDOs with one or more deaths.

### 2.2. Diagnostic Criteria

This study mainly analyzed data on chemical FBDOs. The etiology of chemical FBDO should be determined according to the following principles, that is, based on epidemiological characteristics, clinical manifestations, laboratory tests, and expert analysis. The following principles must be met to determine the etiology of chemical FBDOs. First, the cases must conform to the epidemiological characteristics and clinical manifestations of a specific chemical foodborne disease. Second, if the laboratory detects the same chemical from the suspected food and patient samples, the etiology and food can be determined. When diagnosed according to the above principles, some outbreaks may have an unknown food; however, based on epidemiological investigation, clinical manifestations, and expert panel analysis, they can be identified as chemical FBDOs with unidentified food. These outbreaks should also be reported to the FDOSS. Outbreaks that do not meet these criteria are not reported to the FDOSS.

The epidemiological characteristics, clinical manifestations of chemical poisonings, are described in a group of documents about the diagnostic criteria and principles of management for FBDOs of different etiologies, which were issued by the Ministry of Health of the People’s Republic of China in 1996 and have been used in outbreak investigations since then [21]. The detection of chemical substances in food and patient samples is carried out in accordance with the corresponding detection and inspection methods in the national food safety standards, which are stipulated in the above documents. In 2011 and 2012, the Ministry of Health of the People’s Republic of China successively issued the “Norms for Epidemiological Investigation of Food Safety Accidents” [22] and the “Technical Guidelines for Epidemiological Investigation of Food Safety Accidents” [23], which set specific standardized requirements for CDCs at all levels to perform investigations of FBDOs. Each FBDO recorded by the FDOSS was investigated and diagnosed based on the normative documents described above.

### 2.3. Data Sources

From 2011 to 2023, 11 prefecture-level and 90 county-level CDCs in Zhejiang Province passively reported all FBDOs to the FDOSS that met the diagnostic criteria. All FBDOs were investigated, and the data were reported to the FDOSS, according to a unified reporting process. The municipal CDC reviews outbreaks reported by the district and county CDC within its jurisdiction, the provincial CDC reviews outbreaks submitted after the municipal review, and the National Food Safety Risk Assessment Center reviews outbreaks submitted after the provincial CDC review, ensuring data quality through three levels of review. The information collected for each outbreak included the reporting region, date of occurrence, physical setting, etiology, food categories, number of cases/hospitalizations/deaths, contributing factors, and other details. Provincial CDC surveillance system managers can export an Excel sheet from the FDOSS, which contains the information described above.

### 2.4. Statistical Analysis

The data used in this study were downloaded from the FODSS containing the information for each outbreak and were checked to ensure accuracy. WPS Office (Version 10.8.0.6501, Kingsoft Corporation, 2016) was used for the descriptive analysis of the data. R version 4.1.2 was used to perform chi-square tests (χ^2^) and multiple comparisons analysis, and statistical significance was set at *p* < 0.05. The population data of prefecture-level cities in Zhejiang Province were obtained from the statistical yearbook from 2011 to 2023 on the official website of Zhejiang Provincial Bureau of Statistics [24]. The population in 2017 (median year from 2011 to 2023) was used as the denominator to calculate the per capita case rate.

## 3. Results

### 3.1. General Characteristics

From 2011 to 2023, 74 chemical FBDOs were reported through the FDOSS in 11 prefectures of Zhejiang Province, resulting in 461 cases, 209 hospitalizations, and one death (Table 1). Chemical FBDOs accounted for 3.94% of all FBDOs, ranking fourth with confirmed etiologies. The percentages of hospitalized cases in outbreaks caused by different types of chemicals were different (χ^2^ = 1047.9, *p* < 0.001). A multiple comparisons analysis using G-tests showed that the “chemicals” group differs significantly from the others (*p* < 0.01), which proves that its hospitalization rate (45.34%) is the highest. The number of cases per chemical outbreak ranged from 2 to 63, with a mean of 6.2 cases.

### 3.2. Temporal Distribution

Chemical FBDOs have been reported annually for more than a decade, with the highest number of outbreaks and cases reported in 2019 (Table 2). Chemical FBDOs were reported for all 12 months, with the highest number reported in June (12) and the lowest in September (1). We analyzed the complete time series (12 months × 13 years) (Figure 1), but no significant seasonality was found. The only death occurred in 2022.

### 3.3. Regional Distribution

From 2011 to 2023, chemical FBDOs were reported in all prefectures in Zhejiang Province, except Zhoushan (Table 3). Hangzhou (18) reported the highest number of outbreaks, followed by Jinhua (17) and Wenzhou (14). Huzhou (one) and Lishui (one) reported the lowest number of outbreaks. Wenzhou had the largest number of cases (130). The incidence of outbreak-related cases reported by the 10 prefectures differed (χ^2^ = 89.605, *p* < 0.001). Wenzhou had the highest incidence of outbreak-related cases (15.77 per million people). The only death occurred in Jinhua.

### 3.4. Setting

The largest percentage of chemical FBDOs occurred in households (71.62%, 53/74), followed by catering services establishments (27.03%, 20/74) (Table 4). The latter group included restaurants (seven), work canteens (six), street stalls (five), rural banquets (one), and school canteens (one). Households (43.82%, 202/461) accounted for the largest percentage of outbreak-related cases, followed by work canteens (27.11%, 125/461) and restaurants (14.32%, 66/461). The settings with the highest average number of cases per outbreak were work canteens (21), followed by school canteens (18), and rural banquets (12). Restaurants (66.67%) had the highest hospitalization rates, followed by street stalls (57.14%), work canteens (48.80%), and households (42.57%). One death occurred in a household.

### 3.5. Food

The misuse of nitrite as a common flavoring agent (salt or sugar) in food preparation was responsible for most chemical FBDOs (33.78%) and cases (40.56%) (Table 5). Liquor was responsible for 17.57% of chemical FBDOs, followed by pickles (14.86%), and other cooked vegetables (10.81%). The percentages of hospitalized cases in outbreaks caused by different foods were different (χ^2^ = 64.711, *p* < 0.001). The results of the multiple comparisons analysis using G-tests are shown in Table 5. The hospitalization rate caused by grain products was the highest (78.57%, 33/42). One death was caused by eating food that was mistakenly prepared with sodium nitrite as table salt.

### 3.6. Etiology

During the study period, the chemical FBDOs reported in Zhejiang Province were mainly caused by nitrite, lead, pesticides, rodenticides, tung oil, and methanol (Table 6). Outbreaks caused by nitrite accounted for the largest percentage (56.76%), followed by lead (17.57%), and pesticides (10.81%). The largest percentage of cases was nitrite (61.61%), followed by pesticides (12.80%), and lead (9.33%).

Combining physical setting and etiology, chemical FBDOs that occurred in households had the most varied etiologies (Figure 2), including nitrite (27), lead (13), pesticides (6), rodenticides (5), tung oil (1), and methanol (1). Most outbreaks due to nitrite (27/42), pesticides (6/8), and rodenticides (5/6) occurred in households. Chemical FBDOs in restaurants were caused by nitrite (6) and pesticides (1). There were six outbreaks in work canteens, of which three were caused by nitrite and the other three by pesticides, rodenticides, and tung oil. Nitrite caused all chemical FBDOs in street stalls.

### 3.7. Chemical FBDOs Caused by Nitrite

A total of 42 chemical FBDOs caused by nitrite have been reported, resulting in 284 cases, 140 hospitalizations, and one death. With the exceptions of Huzhou and Zhoushan, nine other prefectures reported outbreaks (Figure 3). The number of outbreaks reported in different regions ranged from 1 to 12, with the most reported in Hangzhou (12), followed by Wenzhou (11), Jinhua (6), and Taizhou (5). Nitrite-related outbreaks reported in the remaining prefectures ranged from one to three. Wenzhou had the highest number of cases (105), followed by Ningbo (61). The number of cases in each outbreak caused by nitrite ranged from 2 to 63. Most outbreaks (71.43%) resulted in 5 or fewer cases, but some (7.14%) had more than 20 cases; the largest single outbreak had 63 cases. From 2011 onwards, the annual number of outbreaks due to nitrite exposure ranged from one to seven, with the highest number of outbreaks reported in 2019 and 2023, with seven outbreaks each (Figure 4). The largest number of cases (107) and hospitalizations (40) occurred in 2019. Only one nitrite-related outbreak occurred in 2022, resulting in one death. Most outbreaks occurred in June (11), and no outbreaks were reported in September (Figure 5). The number of reported outbreaks in other months ranged from one to five. Only three outbreaks were reported in December, but they caused the highest number of cases, with one outbreak resulting in 63 cases, 31 hospitalizations, but no deaths.

Outbreaks caused by nitrite occurred mainly in households (27), followed by restaurants (6), street stalls (5), and work canteens (3) (Figure 2). Among all nitrite-related outbreaks, 59.52% (25/42) were caused by cooking food with nitrite as a common seasoning, 26.19% (11/42) by eating pickled vegetables, 7.14% (3/42) by eating cooked meat products, and 4.76% (2/42) by eating grain products. One outbreak was caused by drinking boiled water. Outbreaks caused by the misuse of nitrite for cooking mainly occurred in households (68%, 17/25), street stalls (16%, 4/25), work canteens (8%, 2/25), and restaurants (8%, 2/25). Outbreaks caused by eating pickled vegetables occurred mainly in households (90.91%, 10/11), and one outbreak occurred in a work canteen. Three outbreaks caused by eating cooked meat products occurred in restaurants (two) and street stalls (one), all of which were caused by the addition of excessive nitrite as a food additive. Two outbreaks caused by eating grain products occurred in restaurants, and investigations found high levels of nitrite in flour products.

### 3.8. Chemical FBDOs Caused by Lead

During the study period, there were 13 chemical FBDOs caused by lead, resulting in 43 cases, 20 hospitalizations, and no deaths. These outbreaks all occurred in households. Liquor was involved in 12 of the outbreaks. They were caused by individuals using a “tin pot” container to hold yellow rice wine for sacrifice, after which the wine left in the container was consumed by the owner or used as a condiment for cooking. These outbreaks occurred only in Jinhua (8) and Taizhou (4). Another lead-related outbreak, caused by drinking water from a new teapot, occurred in Hangzhou. Outbreaks caused by lead have been reported since 2018, with four each in 2020 and 2021, three in 2018, and one each in 2019 and 2023. The outbreaks occurred mainly in the first half of the year, with one, two, three, one, and four outbreaks reported from January to May, respectively; two outbreaks were reported in July, and no outbreaks reported in the other months.

### 3.9. Chemical FBDOs Caused by Other Etiologies

Eight chemical FBDOs caused by pesticides resulted in 59 cases, 30 hospitalizations, and no deaths. Outbreaks occurred in 2013 (one), 2014 (one), 2016 (three), 2019 (one), 2020 (one), and 2021 (one). Except for January, March, September, and December, one outbreak was reported in each month. The foods involved in the outbreaks were vegetables (seven) and fruits (one). Pesticide-related outbreaks were caused by herbicides (four) and insecticides (four). Outbreaks caused by pesticides occurred mainly in households (six), restaurants (one), and work canteens (one) (Figure 2). They occurred in Ningbo (three), Hangzhou (one), Huzhou (one), Shaoxing (one), Jinhua (one), and Wenzhou (one).

The six rodenticide-related outbreaks resulted in 59 cases, 20 hospitalizations, and no deaths. These outbreaks occurred in 2015 (two), 2016 (one), 2021 (one), 2022 (one), and 2023 (one) in January (one), March (two), April (one), August (one), and October (one) in Hangzhou (three), Wenzhou (one), Jinhua (one), and Quzhou (one). The contaminated foods included grain products (two), meat (two), vegetables (one), and unidentified food (one). One of these outbreaks was caused by the consumption of meat from dogs poisoned with rodenticides. These outbreaks occurred mainly in households (five), with only one occurring in a work canteen (Figure 2). The outbreak in the work canteen was caused by the serving of pig lungs mixed with rodenticides to workers. The rodenticides responsible for these outbreaks were bromadiolone (three) and brodifacoum (two), and one outbreak involved both.

Three tung oil-related outbreaks caused 35 cases with two hospitalizations and no deaths. The outbreaks occurred in December 2012, September 2016, and February 2019 in Hangzhou (one), Jiaxing (one), and Quzhou (one) in a household (one), work canteen (one), and rural banquet (one), all of which were caused by preparing food using tung oil as the cooking oil (Figure 2). In addition, there was a methanol-related outbreak in Jinhua in 2018 caused by homemade liquor consumed in a household. Another outbreak in Wenzhou in 2012 was caused by rancid oil in a rural school canteen. No outbreaks were reported in other years.

## 4. Discussion

Since FBDO surveillance began in 2011, chemical FBDOs have been reported every year in Zhejiang Province; there has been no obvious downward trend, and one death occurred in 2022. Therefore, chemical FBDOs require continuous attention. The leading cause of chemical FBDOs in Zhejiang Province is nitrite, with 56.76% of chemical outbreaks being caused by nitrite. Other studies have also shown that nitrite poisoning is the main cause of chemical FBDOs, such as in Gansu (45%) [25], Anhui (86%) [26], and mainland China [27]. 

Nitrite is a general term for a class of inorganic compounds, primarily sodium nitrite (NaNO_2_). Sodium nitrite is a salty white or yellow powder that is similar in appearance and taste to salt, highly water-soluble, and widely used in industrial chemistry, pharmaceutical production, food processing, and other fields [28,29]. Furthermore, it is also a therapeutic agent for cyanide poisoning. The main toxic mechanism of sodium nitrite is the oxidation of ferrous iron (Fe^2+^) in hemoglobin to iron-containing (Fe^3+^) methemoglobin (MetHb), which loses its ability to bind and transport oxygen, resulting in methemoglobinemia [30]. In addition, nitrite acts as a potent vasodilator in peripheral blood vessels, inducing vasodilatory shock [31]. The reported lethal dose range of sodium nitrite is between 0.7 and 6 g [32]. Based on typical doses for treating adults with sodium nitrite for cyanide poisoning, the lethal dose is approximately 2.6 g [33].

Nitrite poisoning is often accidental, and one of the most common causes of poisoning is it being mistaken for kitchen salt or sugar in food preparation [34]. For example, in 2013, a fatal incident of sodium nitrite poisoning occurred on a ship because the cook mistakenly added a spoonful of sodium nitrite to the breakfast prepared for everyone [35]. In another outbreak of nitrite poisoning in China, all the members of a family were poisoned by asparagus fried with nitrite that was mistaken for sugar [36]. In many similar outbreaks [35,36,37,38,39], nitrite was improperly stored, carelessly left in the kitchen, not clearly marked, or incorrectly marked. As nitrite is similar in appearance to table salt and sugar, it can be easily used as a common seasoning to cook food and cause poisoning. The mistaken identification of nitrite as salt or sugar was responsible for 74.29% of the outbreaks in Zhejiang Province, including the one death. This is also the main cause of nitrite poisoning in other provinces of China such as Sichuan [40] and Anhui [26]. Therefore, it is recommended that consumers purchase salt from regular commercial outlets and do not use “salt” substances of unknown origin. Families and individuals should store nitrite in places that are strictly isolated from food, cooking utensils, and drinking water, and containers containing nitrite should be clearly labeled to avoid misuse. Another cause of sodium nitrite poisoning is its excessive use as a food additive. Sodium nitrite is an additive that has multiple functions in meat products. It not only gives meat a bright color and unique flavor [41,42,43], but it also has a strong antibacterial effect, particularly against *Clostridium botulinum* [44]; therefore, it is widely used in meat processing. Our study identified two outbreaks caused by the illegal use of nitrite by food service providers, with excess nitrite added to chicken legs and duck heads. In addition, outbreaks caused by the inappropriate use of nitrite as a food additive by individuals in households have also been reported. In 2018, for example, three people were poisoned after eating homemade sausages given by neighbors, one of whom died. An investigation found that the concentration of sodium nitrite in the homemade sausages was almost 30 times higher than the concentration permitted by local law [45]. In many countries, large quantities of highly purified nitrites are available to consumers for use in preserving meat and other food preservation applications. Sodium nitrite is sold online at a low price and is readily available to consumers, which may increase the risk of individuals potentially ingesting fatal doses [32]. Therefore, some countries have introduced laws and regulations that restrict the sale and use of nitrite. For example, in Australia, sodium nitrite and potassium nitrite are permitted for use in the preparation of processed meat, poultry, and game products, either as preservatives or to improve appearance; however, their use in home cooking is prohibited [46]. In China, the announcement of the State Administration for Market Regulation on the prohibition of nitrite for food service providers and the strengthening of regulations for alcohol-based fuel management (No.18, 2018) [47] stipulates that food service providers are prohibited from purchasing, storing, and using nitrite (including sodium nitrite and potassium nitrite) to avoid mistakenly using nitrite as salt in processed food. The announcement was made on 6 February 2018, but nitrite-related outbreaks still occurred at restaurants in 2019. The outbreaks of the addition of excessive nitrite to chicken legs (March 2019) and duck heads (April 2019) were also caused by improper procurement, storage, and the use of nitrite by catering service providers. It is recommended that the government should intervene and monitor the sodium nitrite market to prevent poisoning caused by misuse and overdose. In addition, it is necessary to further strengthen regulations for the supervision of catering service providers, increase the intensity of inspections and punishment, and reduce the occurrence of such outbreaks.

In addition to poisoning caused by accidental ingestion and the illegal use of sodium nitrite as a food additive, this study found that the consumption of pickled vegetables was responsible for 26.19% (11/42) of nitrite-related outbreaks, and these outbreaks mostly occurred in households (90.91%, 10/11). Pickled vegetables have a long history of use; their processing methods are simple, they are inexpensive, easy to preserve, and have a unique flavor that is popular among consumers. However, pickling vegetables reduces the nitrate naturally present in fresh vegetables to nitrite by the action of bacteria that produce nitrate reductase during the pickling process. The accidental consumption of pickled vegetables with a high nitrite content may lead to poisoning. The nitrite content in pickled vegetables is affected by many factors, such as the nitrate content in fresh vegetables, the freshness of the vegetables, temperature, salinity, and pickling time [48,49]. Studies have reported that the nitrate content of different vegetable varieties varies, and leafy vegetables are more likely to accumulate nitrate than root vegetables, melons, and legumes [50,51]. The nitrate content of the same variety of vegetables is also affected by nitrogen fertilizer, harvest time, temperature, light, and soil characteristics [49]. In addition, the nitrite concentration in fresh and undamaged vegetables is usually low, and as the storage time of fresh vegetables increases, the nitrate content increases owing to bacterial contamination and the action of endogenous nitrate reductase [52]. Moreover, the higher the temperature and the longer the storage time, the higher the nitrite and nitrate content in leafy vegetables [53]. Therefore, fresh vegetables should be used as raw materials when pickling vegetables at home, and they should be kept at a low temperature before pickling. As for the temperature, time, and salinity of pickling, taking the Chinese traditional pickled vegetable potherb mustard as an example, a study reported that it is best eaten after 30 days of pickling at a temperature of 20 °C with a salt concentration between 10% and 15% [54]. In addition, 218 types of packaged and unpackaged pickles from China were tested [55] and the nitrite content of packaged pickles was significantly lower than that of unpackaged pickles. Therefore, we recommend the consumption of packaged pickles rather than unpackaged pickles to reduce nitrite exposure. In addition, it is worth noting that 6 of the 11 pickled vegetable outbreaks occurred in June, and these are also responsible for the peak in the chemical outbreaks in June. Summer (June to August) is a period of strong growth of amaranth; at this time, it is easy to obtain fresh materials for pickling and the summer temperature is high, which is conducive to the fermentation process in the process of pickling. Therefore, pickling at this part of the year is relatively common, and improper pickling can easily to lead to poisoning. Therefore, it is recommended that the relevant departments issue a warning about the prevention of such poisoning in late spring and early summer.

Lead was the second leading cause of chemical FDBOs in Zhejiang Province, and 92.31% (12/13) of the lead outbreaks were caused by the consumption of yellow rice wine in tin cans. Lead poisoning has also been reported in other provinces in China [56,57]. Lead poisoning caused by yellow rice wine in tin pots has regional characteristics; Jinhua and Taizhou in Zhejiang Province have high incidences. Yellow rice wine is traditionally poured into tin pots and then into cups or sprinkled on the ground to honor ancestors [58]. After the sacrifice ceremony, the remaining wine in the tin pot is used as cooking wine or directly consumed. When the wine has been consumed, the tin pot is stored until the next sacrifice. Occasionally, however, the tin pot continues to serve as a container for yellow rice wine or as a container for heating wine. However, because these “tin pots” used for sacrifice contain lead, and since the pH of yellow rice wine is 3.5–4.6, according to the national standard [59], when yellow rice wine is held in tin pots, the organic acids in the yellow rice wine dissolve the lead in the tin pot. The dissolution rate accelerates with increased temperature [60]; thus, those who consume yellow rice wine prepared in this manner are prone to lead poisoning. It is suggested that relevant departments should increase popular science publicity in high-incidence areas. WeChat, short videos, posters, television, official media, and village radio stations can be used to raise awareness of lead poisoning caused by tin pots of wine, and people can be encouraged to only use tin pots as sacrificial or ornamental objects, not as containers for wine and hot wine. The importance of using food-grade materials to prepare and store food should be emphasized. It is also recommended that the relevant departments introduce labeling regulation and clearly indicate that the tin pot used for sacrifice is a sacrificial product that cannot be used to contain food. In addition, local medical personnel should increase their awareness of bioactive lead poisoning, understand the potential sources of lead in the region, and promptly identify and treat cases of lead poisoning. If individuals discover that they have consumed wine from a tin pot, it is recommended that they urgently proceed to a hospital for a blood lead examination and not delay diagnosis and treatment because of the absence of symptoms.

There are little accumulated data on FBDOs caused by other chemicals. Pesticide-related outbreaks are primarily caused by the consumption of contaminated vegetables or fruits. Some vegetables and fruits that have recently been sprayed with pesticides are eaten by unsuspecting individuals. In response to such outbreaks, publicity regarding the dangers of pesticide poisoning should be heightened, and warning signs should be placed where pesticides have been sprayed. In particular, the regulations for management of public environmental weeding departments should be strengthened. They should post notices when spraying pesticides and fulfill their obligations regarding notification. Individuals should also avoid direct consumption of wild fruits that can be contaminated with pesticides. Pesticide poisoning in catering services establishments suggests that effective systems should be established to improve the quality inspection of raw materials. Relevant departments should also pay attention to vegetables and fruits with excessive pesticides sold in the circulation link, and targeted supervision should be strengthened. Pesticide users should regulate their use, such as not using banned pesticides, and should use them strictly in accordance with the pesticide label. The labeled scope of use, method of use, dose, technical requirements, and precautions should be used. Pesticides labeled with the safety interval period should be stopped before the harvest of agricultural products in accordance with the requirements of the safety interval period [61]. Rodenticide poisonings were caused by the ingestion of food contaminated with bromadiolone or brodifacoum, and most of these outbreaks occurred in households. When using rodenticides at home, it is recommended to consult a professional to ensure that the most suitable product is chosen and that the product instructions have been read before use and carefully followed. It is recommended that rodenticides are stored in a designated locked cabinet and attention should be paid to ensure they are not in the vicinity of food and drinks to prevent accidental ingestion. It is recommended that rodenticides are not randomly applied, but a specific poison bait box should be used that only rats can enter. In addition, regular observation, cleaning, and the correct disposal of waste are required. In addition to the household outbreak, an outbreak occurred in a workplace canteen, caused by the canteen staff mistakenly serving pig lungs mixed with rodenticides to workers. The investigation found that the causes of the outbreak were as follows: first, there was no warning color or toxicity sign on the packaging of the poisonous pig lungs; second, there was no independent rodent killing bait-releasing area in the factory; third, there was no record of rodent bait use or placement in the factory, and no notification was provided after bait placement. Therefore, to avoid such outbreaks, it is recommended that work units establish a system of dedicated personnel and a standardized management of rodenticides, establish and standardize records of rodenticide use and delivery, establish rodent bait delivery areas, and standardize the setting of warning colors and toxicity signs. If the work unit is not capable of rat eradication, hiring a third-party professional organization to assist is recommended. Three other FBDOs were caused by tung oil, all of which were caused by the misuse of tung oil as a vegetable oil for cooking. Tung oil is extracted from tung seeds as raw material, which is the main raw material for the manufacture of paint and ink. Tung oil is more toxic, and its main toxic components are ketoacids. In recent years, sporadic outbreaks of poisoning caused by the consumption of tung oil and tung oil seeds have occurred. The appearance of tung oil is similar to that of edible vegetable oil, and tung oil sold in the market is mostly in bulk, without any dangerous goods information. Owing to poor risk awareness of buyers and sellers, tung oil is often decanted into empty edible oil bottles, mineral water bottles, and other containers for convenience in the sale process, which carries the risk of poisoning caused by its use as an edible oil [62]. These outbreaks have occurred in work canteens, small restaurants, families, schools, and other places [63,64,65]. In response to these outbreaks, it has been suggested that relevant departments increase publicity and improve education, inform residents to correctly store and label tung oil, and not use containers that have been exposed to tung oil to hold cooking oil. Cooking oil and tung oil should be kept separately; to prevent the misuse of tung oil, it should not be stored in the kitchen. In addition, it is necessary to strengthen the regulation for the management of tung oil and other industrial oils, and packaging should have clear labeling. Relevant departments should improve food monitoring and supervision of construction site canteens, rural banquets, and other key places.

This study had certain limitations. Although our surveillance system has greatly increased outbreak reporting, underreporting has inevitably occurred for several reasons. For example, if patients do not cooperate, epidemiological investigations cannot be performed efficiently and samples cannot be collected. In addition, in some cases, testing has not been able to identify toxins, the investigations of some outbreaks have been incomplete, and the causes and foods are unknown. Our study drew more conclusions from confirmed outbreaks; however, these conclusions may not fully reflect the overall status of chemical FBDOs. Only a small amount of data were collected for outbreaks caused by some chemicals, which could affect the proposed targeted measures.

## 5. Conclusions

In view of the frequent outbreaks of chemical foodborne diseases in our province from 2011 to 2023, a variety of prevention and control measures were proposed based on the research results of the temporal and regional distribution, foods and food establishments involved, and etiological agents of chemical FBDOs. However, the effectiveness of these recommendations needs to be further verified and studied. In general, public health institutions should further strengthen the surveillance and health education of the population. Individuals should store toxic chemicals, such as nitrates, pesticides, and rodenticides correctly to avoid poisoning by ingestion. In view of the chemical FBDOs caused by food in the catering and distribution links, relevant departments should strengthen targeted supervision. In the future, the effectiveness of the above measures can be further studied to better promote food safety practice and policy making.

## Figures and Tables

**Figure 1 foods-14-00936-f001:**
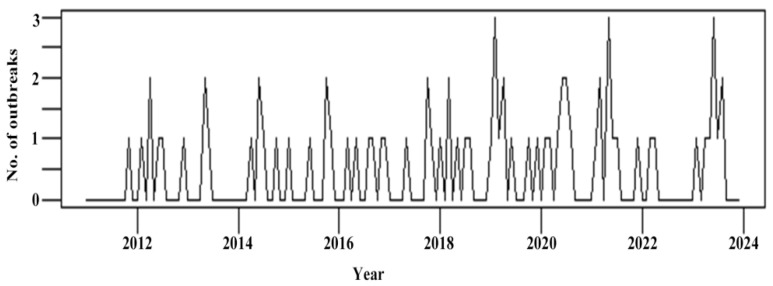
Time series of chemical foodborne disease outbreaks in Zhejiang Province, 2011–2023.

**Figure 2 foods-14-00936-f002:**
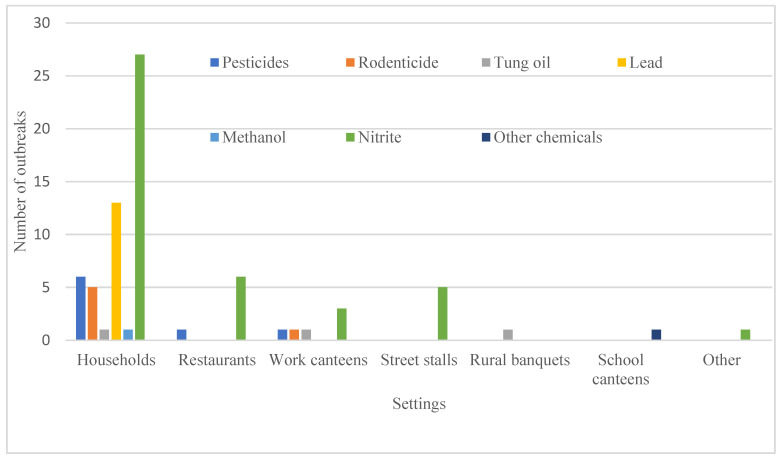
Setting etiology distribution of chemical foodborne disease outbreaks in Zhejiang Province, 2011–2023.

**Figure 3 foods-14-00936-f003:**
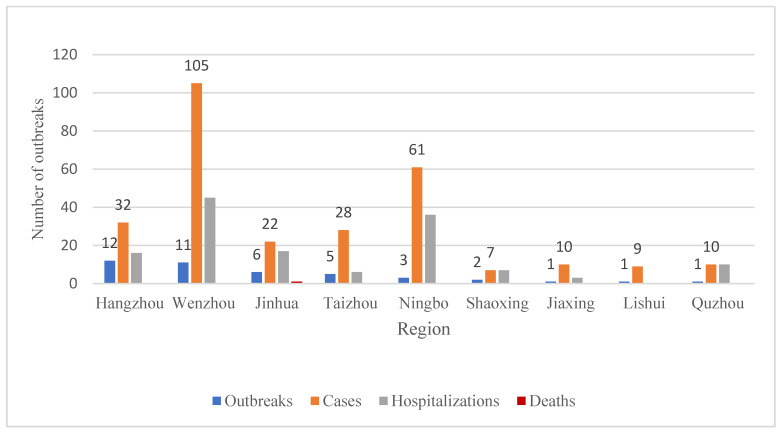
Regional distribution of foodborne disease outbreaks caused by nitrite, Zhejiang Province, 2011–2023.

**Figure 4 foods-14-00936-f004:**
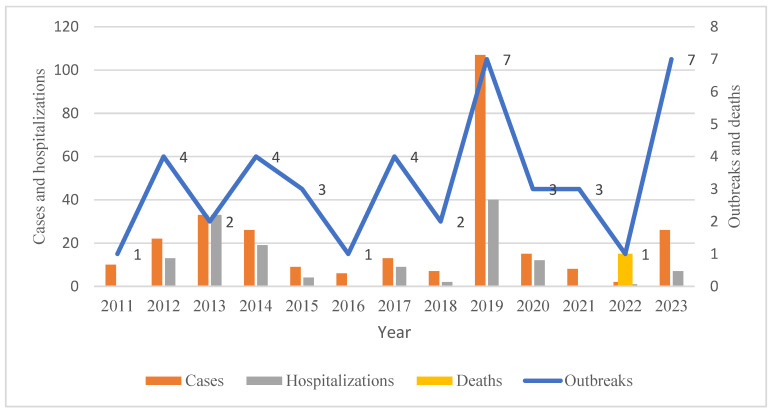
Annual distribution of foodborne disease outbreaks caused by nitrite, Zhejiang Province, 2011–2023.

**Figure 5 foods-14-00936-f005:**
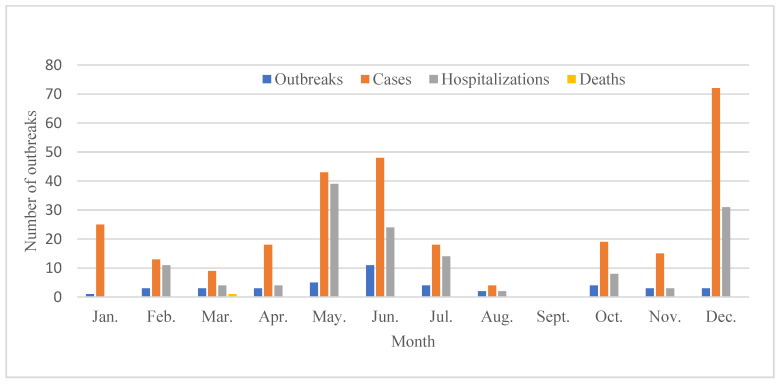
Monthly distribution of foodborne disease outbreaks caused by nitrite in Zhejiang Province, 2010–2023.

**Table 1 foods-14-00936-t001:** Number of foodborne disease outbreaks, cases, hospitalizations, and deaths, by etiology, in Zhejiang Province, 2011–2023.

Etiology	Outbreaks		Cases		Hospitalizations		Deaths		Mean Number of Cases per Outbreak	Hospitalizations /Cases × 100% ^††^	*p*-Value ^†^
	Number	%	Number	%	Number	%	Number	%			
Fungi and their toxins	408	21.73	1408	8.67	403	20.70	20	68.97	3.5	28.62 ^a^	<0.001
Plant toxins	106	5.64	631	3.88	116	5.96	4	13.79	6.0	18.38 ^b^	
Animal toxins	34	1.81	241	1.48	27	1.39	1	3.45	7.1	11.20 ^c^	
Biological *	784	41.75	10,277	63.27	1009	51.82	3	10.34	13.1	9.82 ^c^	
Chemicals **	74	3.94	461	2.84	209	10.73	1	3.45	6.2	45.34 ^d^	
Unknown etiology	469	24.97	3214	19.79	183	9.40	0	0.00	6.9	5.69 ^e^	
Multiple etiologies	3	0.16	11	0.07	0	0.00	0	0.00	3.7	0.00	
Total	1878	100.00	16,243	100.00	1947	100.00	29	100.00	8.6	11.99	

*: “Biological” include bacteria and their toxins and viruses. **: “Chemicals” means the chemical hazards except for the toxins. ^†^: Chi-square test. H0: All rates are equal. ^††^: Pairwise. G-test: a, b, c, … Different superscripts indicate significant differences for *p* < 0.01.

**Table 2 foods-14-00936-t002:** Annual distribution of chemical foodborne disease outbreaks, Zhejiang Province, 2011–2023.

Year	Outbreaks		Cases		Hospitalizations		Hospitalizations/Cases	Deaths
	Number	%	Number	%	Number	%	×100%	
2011	1	1.35	10	2.17	0	0.00	0.00	0
2012	6	8.11	59	12.80	13	6.22	22.03	0
2013	3	4.05	38	8.24	33	15.79	86.84	0
2014	5	6.76	28	6.07	19	9.09	67.86	0
2015	5	6.76	14	3.04	8	3.83	57.14	0
2016	6	8.11	26	5.64	7	3.35	26.92	0
2017	4	5.41	13	2.82	9	4.31	69.23	0
2018	6	8.11	20	4.34	5	2.39	25.00	0
2019	10	13.51	129	27.98	42	20.10	32.56	0
2020	8	10.81	37	8.03	20	9.57	54.05	0
2021	9	12.16	48	10.41	37	17.70	77.08	0
2022	2	2.70	5	1.08	4	1.91	80.00	1
2023	9	12.16	34	7.38	12	5.74	35.29	0
Total	74	100.00	461	100.00	209	100.00	45.34	1

**Table 3 foods-14-00936-t003:** Regional distribution of chemical foodborne disease outbreaks, Zhejiang Province, 2011–2023.

Region	Outbreaks		Cases		Hospitalizations		Deaths	Incidence Rates of Cases per Million	*p*-Value *
	Number	%	Number	%	Number	%	Number		
Hangzhou	18	24.32	68	14.72	25	11.96	0	9.02	<0.001
Jinhua	17	22.97	60	12.99	29	13.88	1	12.36	
Wenzhou	14	18.92	130	28.14	48	22.97	0	15.77	
Taizhou	9	12.16	41	8.87	19	9.09	0	6.79	
Ningbo	6	8.11	73	15.80	36	17.22	0	12.23	
Quzhou	3	4.05	27	5.84	17	8.13	0	10.47	
Shaoxing	3	4.05	14	3.03	7	3.35	0	3.14	
Jiaxing	2	2.70	14	3.03	3	1.44	0	3.93	
Huzhou	1	1.35	25	5.41	25	11.96	0	9.39	
Lishui	1	1.35	9	1.95	0	0.00	0	3.34	
Total	74	100.00	461	99.78	209	100.00	1	9.48	

*: Chi-square test. H0: All rates are equal.

**Table 4 foods-14-00936-t004:** Number and percentage of chemical foodborne disease outbreaks, cases, hospitalizations, and deaths, by setting, Zhejiang Province, 2011–2023.

Setting	Outbreaks		Cases		Hospitalizations		Deaths	Hospitalizations/Cases ×100%	Mean Number of Cases per Outbreak
	Number	%	Number	%	Number	%	Number		
Households	53	71.62	202	43.82	86	41.15	1	42.57	4
Restaurants	7	9.46	66	14.32	44	21.05	0	66.67	9
Work canteens	6	8.11	125	27.11	61	29.19	0	48.80	21
Street stalls	5	6.76	28	6.07	16	7.66	0	57.14	6
Rural banquets	1	1.35	12	2.60	2	0.96	0	16.67	12
School canteens	1	1.35	18	3.90	0	0.00	0	0.00	18
Other	1	1.35	10	2.17	0	0.00	0	0.00	10
Total	74	100.00	461	100.00	209	100.00	1	45.34	6

**Table 5 foods-14-00936-t005:** Number and percentage of chemical foodborne disease outbreaks, cases, hospitalizations, and deaths, by food, Zhejiang Province, 2011–2023.

Food	Outbreaks		Cases		Hospitalizations		Deaths Number	100 × Hospitalizations/Cases (%)	*p*-Value **
	Number	%	Number	%	Number	%			
Non-food (nitrite)	25	33.78	187	40.56	95	45.45	1	50.80 ^a^	<0.001
Liquor	13	17.57	43	9.33	19	9.09	0	44.19 ^a, d^	
Vegetable (pickle)	11	14.86	30	6.51	11	5.26	0	36.67 ^a, d^	
Other cooking vegetables	8	10.81	57	12.36	30	14.35	0	52.63 ^a^	
Grain product	4	5.41	42	9.11	33	15.79	0	78.57 ^b^	
Cooked meat	5	6.76	30	6.51	13	6.22	0	43.33 ^a, d^	
Non-food (Tung oil)	3	4.05	35	7.59	2	0.96	0	5.71 ^c^	
Water	2	2.70	12	2.60	2	0.96	0	16.67 ^d, e, c^	
Fruit	1	1.35	5	1.08	3	1.44	0	60.00 ^a, b, e^	
Vegetable oil	1	1.35	18	3.90	0	0.00	0	0.00 ^c^	
Unidentified food	1	1.35	2	0.43	1	0.48	0	50.00	
Total	74	100.00	461	100.00	209	100.00	1	45.34	

(**) Chi-square test for simultaneous group comparison. Multiple comparisons: different superscripts indicate significant differences for *p* < 0.05. There is a significant difference (*p* < 0.05) between two groups when they do not share the same letter.

**Table 6 foods-14-00936-t006:** Number and percentage of chemical foodborne disease outbreaks, cases, hospitalizations, and deaths, by etiology, Zhejiang Province, 2011–2023.

Etiology	Outbreaks		Cases		Hospitalizations		Deaths Number
	Number	%	Number	%	Number	%	
Nitrite	42	56.76	284	61.61	140	66.99	1
Lead	13	17.57	43	9.33	20	9.57	0
Pesticides	8	10.81	59	12.80	30	14.35	0
Rodenticides	6	8.11	20	4.34	16	7.66	0
Tung oil	3	4.05	35	7.59	2	0.96	0
Methanol	1	1.35	2	0.43	1	0.48	0
Other chemicals	1	1.35	18	3.90	0	0.00	0
Total	74	100.00	461	100.00	209	100.00	1

## Data Availability

The study of the data can be downloaded from the Foodborne Disease Outbreaks Surveillance System (https://sppt.cfsa.net.cn/goto (accessed on 1 June 2021)), and the data are not currently open.

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
