# Peer review of "Surveillance of Chemical Foodborne Disease Outbreaks in Zhejiang Province, China, 2011–2023"

_foods, 2025, doi:10.3390/foods14060936_

Round 1
Reviewer 1 Report
Comments and Suggestions for Authors
Title : Surveillance of chemical foodborne disease outbreaks in Zhejiang Province, China, 2011-2023
In this manuscript, data of chemical foodborne disease outbreaks in Zhejiang Province between 2011-2023 were analyzed. The outbreaks by chemical agents are important in food safety area, but the more comprehensive comparison of the data with previous research should be conducted in the discussion section. Specific comments are as follows;
Abstract: The described abstract is too long. Please revise it more concisely with main results
L 51-52 : Reference is needed for this sentence
L 120 : The period of study (2011 to 2022) is not same with the title (2011-2023)
L 134 : Since when?
L 177 : The analysis from the population data of each year would be more powerful rather than the median value. The annual data of population was not obtained?
Table 1 : The classification of the etiology seems not clear. “Biological” means the outbreaks from the bacteria and virus? “Chemicals” means the chemical hazards except for the toxins? Please clarify the type of outbreaks in Table 1.
Figure 1: Even though the author insisted that the no obvious seasonal characteristics was observed, the difference between the outbreaks in June and September was significant. Therefore, it should be more discussed with other references. Also in Figure 5, specific analysis with the nitrite cases showed that the outbreaks were focused on the June rather than September.
L 199-208 : It seems that the combined description of Results & Discussion is more appropriate in this manuscript because the interpretation of data has limited meaning. What is the meaning of the regional distribution in this section?
L 209 : Please revise the word “physical settings” with “Places” or other appropriate expression.
L 303 : Please add figure (Figure 6) and revise Figure 5 as Figure 6 in the figure legend
L 349-394 : As describe in this sentences, nitrite poisoning is important chemical outbreaks. It is recommended to compare the nitrite poisoning statistics in Zhejiang Province with other regions with references.
Reviewer 2 Report
Comments and Suggestions for Authors
Chen et al. conducted a study titled Surveillance of Chemical Foodborne Disease Outbreaks in Zhejiang Province, China, 2011–2023, which analyzed the epidemiological characteristics of chemical foodborne disease outbreaks (FBDOs) in Zhejiang Province over a specified period (2011–2023). By examining and analyzing the number of outbreaks, cases, hospitalizations, and deaths, the study highlights the significance of outbreaks caused by chemical agents compared to those of other origins, as well as the impact of factors such as temporal and regional distribution, type of food establishment, etiological agents, and the foods involved. The study concluded that chemical FBDOs resulted in the highest percentage of hospitalized cases and did not exhibit clear seasonal or regional patterns. Households and catering services were the most frequently implicated establishments, with nitrite, lead, pesticides, rodenticides, tung oil, and methanol identified as the primary causative agents. For outbreaks caused by nitrite and lead—deemed the most critical—the study provided detailed descriptions of the establishments, responsible foods, and regions with the highest incidence. Similar details were reported for other chemical agents examined in the study. Finally, to propose targeted prevention and control measures, the authors recommend food safety training for the general population, provided by public health institutions. This training should focus on the proper storage of toxic chemicals and the implementation of stricter supervision standards for catering services and food distribution systems.
The study exhibits certain flaws in data analysis that may result in erroneous or ambiguous interpretations. Nevertheless, I acknowledge the significance of the proposed approach and objective, as well as the value of the database utilized and its implications for consumer health. Therefore, I recommend that the article undergo major revision.
To enhance the writing, quality, and clarity of the article, we offer the following comments for the authors’ consideration:
1. In the “Methods” section, it would be beneficial to clearly describe the criteria used to evaluate the epidemiological characteristics, clinical manifestations, and laboratory threshold levels detected for each chemical agent studied.
2. “Figure 5” is duplicated, appearing as both a graph and a photo. Please revise and clarify this discrepancy to ensure the content is accurately presented.
3. In the “Discussion” section, the control measures recommended for each chemical agent are expected to reduce outbreaks and cases. However, it is unclear whether there is supporting evidence from other authors demonstrating the efficacy of these measures or if they are merely suggestions without validated outcomes. Please clarify whether these recommendations are evidence-based or still theoretical.
4. In the “Conclusions” section, it is advisable to avoid repeating the results already discussed. Instead, focus on the broader significance of the findings and their potential impact on food safety practices and policies.
5. In the “Reference” section, please address the following issues:
• Some links lead to a “DOI not found” error (e.g., references 19, 41, 43, 50…). It would be helpful to verify and correct these references.
• Certain references are in Chinese (e.g., 22, 23, 59, 61…), making them inaccessible for review. If possible, provide an English translation or a brief summary of these sources.
Additionally, here are some considerations regarding the analysis of the data:
6. Lines 181-186. According to the authors, the value of the chi-square test statistic is 1047.0. This value is based on the following contingency table:
|
Fungi |
Plant |
Animal |
Biological |
Chemicals |
Unknown |
Multiple |
Hosp |
403 |
116 |
27 |
1009 |
209 |
183 |
0 |
No hosp |
1005 |
515 |
214 |
9268 |
252 |
3031 |
11 |
The chi-square test for the comparison of hospitalization rates (H0: All rates are equal) is summarized as:
## Pearson's Chi-squared test
##
## X-squared = 1047.9, df = 6, p-value < 2.2e-16
What the significance of the test indicates is that the rates of hospitalizations between groups are not homogeneous. It does not follow that one of them is the highest. The next step should be to identify by a multiple comparisons method the pairs of rates that are significantly different. From these comparisons it can be concluded that the chemical group has the highest rate of hospitalizations. For multiple comparisons, the authors could use the “pairwise.G.test” function corresponding to the R package “RVAideMemoire”. This procedure yields a matrix of p-values corresponding to the comparisons of all pairs of rates. For the current data, the matrix obtained is:
Pairwise comparisons using G-tests
data: t(tb)
Fungi Plant Animal Biological Chemicals Unknown
Plant 9.6e-07 - - - - -
Animal 1.8e-09 0.01077 - - - -
Biological < 2e-16 6.1e-10 0.48396 - - -
Chemicals 1.5e-10 < 2e-16 < 2e-16 < 2e-16 - -
Unknown < 2e-16 < 2e-16 0.00254 1.5e-13 < 2e-16 -
Multiple 0.00921 0.04371 0.12846 0.14558 0.00048 0.26934
Note that the “chemicals” group differs significantly from the others, which proves that its hospitalization rate is the highest.
7. Lines 189-193. I don't know what the authors mean by seasonality of a time series. I guess Figure 1 shows for each of the four series (outbreaks, cases, hospitalization, deaths) the sum of frequencies in each month over the 13 years of observations. Suppose now that the profiles for each year (13) had been similar to the one shown in Figure 1 with a peak in June and a trough in September. Would the authors then say that there is no seasonality? Please clarify.
8. In relation to the previous point, the authors should have considered the complete time series of events (12 months x 13 years) to perform an analysis to extract the trend and the eventual seasonality component.
9. In Figure 1, the labels of the ordinate axes are confusing. The expression “cases/hospitalizations” could be confused with the quotient of both variables. It would be better to use “cases and hospitalizations”. Idem with “number of outbreaks/death”.
10. I think the data shown in Table 5 should be further analyzed. I suggest performing the same analysis as described in point 1.
11. These data are insufficiently explored. In future research, the authors could consider multidimensional spatio-temporal count models that would allow obtaining adjusted rates.
Implementing these changes could enhance the study’s reliability, potentially necessitating revisions to the corresponding sections of the Abstract, Results, Discussion, and Conclusions.
Round 2
Reviewer 1 Report
Comments and Suggestions for Authors
The authors revised manuscript significantly considering the reviewers comments, but it is recommended to replace or delete the Figure 7. Also, check the manscript one more time and revise it more clearely.
Author Response
The authors revised manuscript significantly considering the reviewers comments, but it is recommended to replace or delete the Figure 7. Also, check the manscript one more time and revise it more clearely.
Response: Thanks for your suggestion, we have removed Figure 7 and thoroughly checked and revised the manuscript.
Reviewer 2 Report
Comments and Suggestions for Authors
Thank you to the authors for their efforts. They have improved the content of the article after the revision, but some aspects still need to be clarified, so I recommend a minor revision. Below, I provide the relevant comments following the points outlined in the original cover letter:
1. I agree with the changes made to the text of the document. The criteria considered for the cases are now clearer. However, since references 21 to 24 are in Chinese and may be difficult to consult for those unfamiliar with the language, the text should at least specify the full identification of the source where they are cited (e.g., Government of China, and if applicable, the specific agency or department).
2. Correct change.
3. Correct. I notice that this observation has been included in the conclusion of the article.
4. Correct, but please take note of the following:
• Line 582: Specify that it refers to the period 2011–2023.
• Line 583: Indicate that the results include studies on the temporal and regional distribution, foods and food establishments involved, and etiological agents of chemical FBDOs.
5. The URL provided for reference 19 is still causing access issues. Please review it again or, alternatively, check if this one might be useful:
https://books.google.es/books?hl=es&lr=&id=MJRbMtPJLLUC&oi=fnd&pg=PP2&dq=World+Health+Organization,+2008.+Foodborne+disease+outbreaks+:+guidelines+for+investigation+and+control.+World+686+Health+Organization.+Foodborne+disease+outbreaks:+guidelines+for+investigation+and+control&ots=Zgzz3sL2qc&sig=Yf4Mgowf8GZrRYX04kCkDRROsMI#v=onepage&q&f=false
The same issue occurs with the current reference 44:
chrome-extension://efaidnbmnnnibpcajpcglclefindmkaj/https://edisciplinas.usp.br/pluginfile.php/5331630/mod_resource/content/1/cura%20ingles.pdf
And with the current reference 50
https://www.cabidigitallibrary.org/doi/full/10.5555/20210007851
I must mention that there are still references that are not accessible with the information provided by the authors (for example, reference 25, which still shows an incorrect DOI, and references 26, 40, etc...). Please review.
Regarding points (6 to 11) mentioned in the cover letter about the data analysis, I would like to comment as follows:
a) Table 1. “Multiple etiologies” are not included in the “Total”. Please, clarify.
b) The authors do not incorporate multiple comparisons analysis in the manuscript, both in the statistical analysis subsection as in the results.
The percentages of hospitalizations according to origin shown in Table 1 would be as follows.
|
Fungi |
Plant |
Animal |
Biological |
Chemicals |
Unknown |
P-value* |
Cases (n) |
1408 |
631 |
241 |
10277 |
461 |
3214 |
< 0.001 |
Hospitalizations (n) |
403 |
116 |
27 |
1009 |
209 |
183 |
|
Hosp / Cases (%) ** |
28.62 a |
18.38 b |
11.2 c |
9.82 c |
45.34 d |
5.69 e |
|
(*) Chi-square test : H0: All rates are equals
(**) G-test: a,b,c,.. Diferent superscripts indicate significant differences for P < 0.01
Note that the fact that the chi-square test is significant does not imply that all hospitalization rates are different. This analysis allows us to affirm that the rate of hospitalizations due to products of chemical origin is the highest because it has the highest frequency and is also significantly different from the rest of the rates.
Multiple comparisons are based on the following triangular matrix
> pairwise.G.test(t(tb1))
Pairwise comparisons using G-tests
data: t(tb1)
Fungi Plant Animal Biological Chemicals
Plant 6.8e-07 - - - -
Animal 1.3e-09 0.0088 - - -
Biological < 2e-16 4.3e-10 0.4840 - -
Chemicals 1.1e-10 < 2e-16 < 2e-16 < 2e-16 -
Unknown < 2e-16 < 2e-16 0.0020 1.1e-13 < 2e-16
P value adjustment method: fdr
c) Regarding Figure 1, it may not make sense to keep it. In any case, to avoid confusion, it should be specified that it simply shows the months in which chemical outbreaks occurred during the years covered by the study, and that it is by no means an analysis of the potential seasonality of the outbreaks, which could actually be observed in Figure 2.
On the other hand, if Figure 1 is to be retained, it would be advisable to reference it in line 481. Also, I believe it would be better to switch the order in which Figures 1 and 2 appear, as Figure 2 is the truly important one, and the text of the document (section 3.2) should refer to it first.
Authors should strive to improve Figure 2. This is the one that shows the lack of seasonality. It could be superimposed by months the outbreaks, cases and hospitalizations.
In sections 3.6 and 3.8, “FIGURE 2” appears several times, and I believe the numbering is incorrect. Please review.
d) I think the data shown in Table 5 should be further analyzed. I suggest performing the same analysis as described in point b). Making all possible comparisons greatly enriches the research.
e) Regarding what the authors mention in point 11 of the cover letter, “In future studies, we will continue to dig deeper into our monitoring data by considering multidimensional spatio-temporal counting models or other statistical analysis methods,” I would like to inform that if you need assistance or advice for future studies using multidimensional spatio-temporal models, we can arrange a research collaboration for data analysis. If you are interested, my contact is: esther.sanjuan@ulpgc.es.
Finally, I recommend reviewing and revising the abstract subsection to reflect the changes made in the manuscript following this latest revision.
Author Response
Please check the attachment, thank you!

Round 3
Reviewer 2 Report
Comments and Suggestions for Authors
Thanks to the authors for incorporating the suggested revisions. However, two final issues remain incorrect or insufficiently explained in the data analysis.
Please see the detail in the attached file.
